# AR-Pro: Counterfactual Explanations for Anomaly Repair with Formal Properties

**Xiayan Ji**[*]    **Anton Xue**[*]    **Eric Wong**    **Oleg Sokolsky**    **Insup Lee**
Department of Computer and Information Science
University of Pennsylvania
Philadelphia, PA 19104
`{xjiae,antonxue,exwong,sokolsky,lee}@seas.upenn.edu`

## Abstract

Anomaly detection is widely used for identifying critical errors and suspicious behaviors, but current methods lack interpretability. We leverage common properties of existing methods and recent advances in generative models to introduce counterfactual explanations for anomaly detection. Given an input, we generate its counterfactual as a diffusion-based repair that shows what a non-anomalous version *should have looked like*. A key advantage of this approach is that it enables a domain-independent formal specification of explainability desiderata, offering a unified framework for generating and evaluating explanations. We demonstrate the effectiveness of our anomaly explainability framework, AR-Pro, on vision (MVTec, VisA) and time-series (SWaT, WADI, HAI) anomaly datasets. The code used for the experiments is accessible at: `https://github.com/xjiae/arpro`.

## 1   Introduction

Anomaly detectors measure how much their inputs deviate from an established norm, where too much deviation implies an instance that warrants closer inspection [15, 52]. For example, unusual network traffic may indicate potential malicious attacks that necessitate a review of security logs [16, 30], irregular sensor readings in civil engineering may suggest structural weaknesses [40, 49, 53], and atypical financial transactions point to potential fraud [3, 47]. As a benign occurrence, unexpected anomalies in scientific data can also lead to new insights and discoveries [25, 45].

Although state-of-the-art anomaly detectors are good at catching anomalies, they often rely on black-box models. This opacity undermines reliability: inexperienced users might over-rely on the model without understanding its rationale, while experts may not trust model predictions that are not backed by well-founded explanations. This limitation of common machine learning techniques has led to growing interest in model explainability [12], particularly in domains such as medicine [50] and law [14]. We refer to [36, 44] and the references therein for recent surveys on explainability.

While many anomaly detection methods can localize which parts of the input are anomalous, this may not be a satisfactory explanation, especially when the data is complex. In medicine [20], heart sound recordings [17] and EEG brain data [5] can differ greatly between individuals; in industrial manufacturing, it can be hard to understand subtle defects of PCB for inexperienced workers [73]. Thus, even when the location of the anomaly is known, it may be hard to articulate *why* it is anomalous. In such cases, it can be helpful to ask the **counterfactual** question: "What *should* a non-anomalous version look like?" For example, a doctor might ask what changes in a patient's chest X-ray might improve diagnostic outcomes [58], while a quality assurance engineer may ask what changes would fix the defect [6].

---

[*]Equal contribution.

38th Conference on Neural Information Processing Systems (NeurIPS 2024).

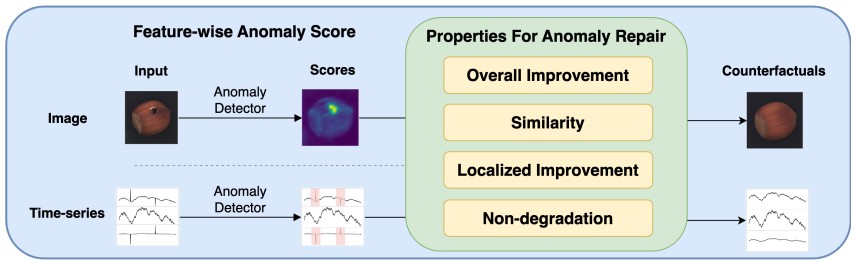

Figure 1: Overview of the AR-Pro framework. We first identify an input's anomalous region and then use property-guided diffusion to repair it. This repair is the counterfactual anomaly explanation, where the following properties are defined with respect to a *linearly decomposable* anomaly detector (AD). (*Overall Improvement*) The repair has a lower anomaly score. (*Similarity*) The repair should resemble the original. (*Localized Improvement*) The score over the repaired region should improve. (*Non-degradation*) The score over the non-anomalous region should not significantly worsen.

To answer such counterfactual questions, we begin with two observations. First, anomalies are commonly localized to small regions of the input [13, 38, 73]. Second, recent generative models, e.g., diffusion [24] models, can be trained to produce high-quality, non-anomalous examples. These observations motivate us to *repair anomalies* as a *counterfactual explanation*. However, simply generating a repair is not sufficient; we must also ensure its quality. For example, it may be undesirable for a repair to significantly improve the anomaly score but barely resemble the original input. Therefore, it is important to measure the quality of repairs using *formal properties*.

However, formalizing broadly applicable properties is challenging, as different domains (e.g., vision, time series) and tasks (e.g., manufacturing, security) have unique considerations. To overcome this, we further observe that many anomaly detectors in practice [4] satisfy a condition that we call *linear decomposability*: the overall anomaly score is an aggregation of feature-wise anomaly scores. Importantly, this is a strong but common condition with which we may formalize the desiderata of counterfactual explanations. Conveniently, many of these desiderata are, in fact, domain-independent. We leverage these conditions to formalize a general, domain-independent framework for counterfactual anomaly explanation: we use a generative model to produce a repair and then evaluate this repair with respect to the detector model.

We present an overview of our anomaly explainability framework, AR-Pro, in Figure 1. While anomaly repair [21, 31, 70] and counterfactual explanations [56] have been explored in the literature, we are the first to study this in a unified context. We summarize our contributions as follows:

- We observe that many anomaly detectors satisfy *linear decomposability* and use this condition to define general, domain-independent properties for counterfactual explanations. This approach lets us measure explanation quality with respect to the anomaly detector, as shown in Figure 1.

- We use these properties to guide diffusion models towards a high-quality repair of the anomalous input. Such a repair serves as the counterfactual explanation of the anomaly.

- Our framework, AR-Pro, can produce semantically meaningful repairs that outperform off-the-shelf diffusion models with respect to our explainability criteria. Our vision anomaly benchmarks include MVTec [13] and VisA [73]. Our time-series anomaly benchmarks include SWaT [38], HAI [54], and WADI [2].

## 2 Overview and Formal Properties

In Section 2.1, we first observe that many anomaly detector paradigms are *linearly decomposable*. Intuitively, this condition means that the overall anomaly score is an aggregation of feature-wise anomaly scores. Next, we use linear decomposability in Section 2.2 to formalize general, domain-independent properties of counterfactual explanations. We will use the terms *counterfactual*, *explanation*, and *repair* interchangeably throughout this paper.

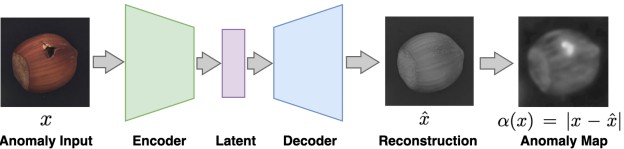

$x$          $\hat{x}$    $\alpha(x) = |x - \hat{x}|$

**Anomaly Input**    **Encoder**    **Latent**    **Decoder**     **Reconstruction**     **Anomaly Map**

Figure 2: Reconstruction-based anomaly detection exemplifies linear decomposition. The anomalous input $x \in \mathbb{R}^n$ is first reconstructed into $\hat{x} \in \mathbb{R}^n$, and the feature-wise anomaly scores are given by $\alpha_i(x) = |\hat{x}_i - x_i|^2 \in \mathbb{R}^n$ for $i = 1, \ldots, n$. Then, the standard $\ell^2$ reconstruction-based anomaly score is a linear combination of the feature scores: $s(x) = \alpha_1(x) + \cdots + \alpha_n(x)$.

## 2.1 Common Anomaly Detectors are Linearly Decomposable

Many anomaly detection techniques use a scoring function $s : \mathbb{R}^n \to \mathbb{R}$ to measure the anomalousness an input $x \in \mathbb{R}^n$, where $s(x)$ is called the *anomaly score* of $x$. This is commonly done in a two-stage process: the *feature-wise anomaly scores* $\alpha_1(x), \ldots, \alpha_n(x) \in \mathbb{R}$ are first computed and then aggregated [15]. We observe that this aggregation often satisfies the following form:

**Definition 2.1** (Linear Decomposition). The feature-wise anomaly scores $\alpha : \mathbb{R}^n \to \mathbb{R}^n$ and regularizer $\beta : \mathbb{R}^n \to \mathbb{R}$ *linearly decomposes* the anomaly score $s : \mathbb{R}^n \to \mathbb{R}$ if for all $x \in \mathbb{R}^n$:

$$s(x) = \alpha_1(x) + \cdots + \alpha_n(x) + \beta(x).$$

While linear decomposability appears to be a strong assumption, it is common in practice. We show a number of examples below, where we also flexibly refer to $\alpha : \mathbb{R}^n \to \mathbb{R}^n$ as the *feature scores*.

**Example 2.2.** In reconstruction-based anomaly detection [8, 29], the input $x$ is passed through an encoder-decoder architecture to generate a *reconstruction* $\hat{x}$. The motivation is that it should be harder to reconstruct out-of-distribution (anomalous) inputs. Empirically, it is observed that if $x$ is anomalous, then $\hat{x}$ will have a large reconstruction error when feature $i$ is in the anomalous region, e.g., a pixel in the defect area of a manufacturing artifact image. A typical example of such an anomaly score is:

$$s(x) = |\hat{x}_1 - x_1|^2 + \cdots + |\hat{x}_n - x_n|^2,$$

which is also known as the $\ell^2$ reconstruction error. This lets us define the feature-wise anomaly scores by $\alpha_i(x) = |\hat{x}_i - x_i|^2$, and we illustrate this example in Figure 2.

**Example 2.3.** In maximum likelihood-based anomaly detection [15], one measures the likelihood of a test input $x$ with respect to a set of non-anomalous training examples. Intuitively, an out-of-distribution (anomalous) $x$ should be unlikely with respect to the non-anomalous training examples and will thus have a lower likelihood. In variants such as normalizing flow-based anomaly detection for images [66], it is common to define a joint probability distribution over all the features:

$$s(x) = -\big[ \log p_1(x) + \cdots + \log p_n(x) + \log|\det J(x)| \big],$$

where $p_1(x), \ldots, p_n(x)$ are the probabilities of each feature that lets us define $\alpha_i(x) = -\log p_i(x)$, while the change-of-variable Jacobian $\log|\det J(x)|$ may be viewed as a regularization term.

**Example 2.4.** In language modeling [61], it is common to measure the anomaly of a token sequence based on the likelihood of each token [7]. Although similar to the vision case, the standard formulation for language models is different: given a token sequence $x_1, \ldots, x_n \in \{1, \ldots, \texttt{vocab\_size}\}$, its measure of unlikeliness (anomalousness) may be defined as:

$$s(x) = -\frac{1}{n}\big[ \log p(x_1) + \log p(x_2|x_1) + \log p(x_3|x_1, x_2) + \cdots + \log p(x_n|x_1, \ldots, x_{n-1}) \big],$$

where $p$ is a probabilistic generative model, and so $\alpha_i(x) = -(1/n) \log p(x_i|x_1, \ldots, x_{i-1})$. This is also known as a *preplexity measure*, which has been used to detect jailbreaks against LLMs [64].

Beyond the above examples, anomaly detectors for time-series data [29, 60, 63] and text [35] also commonly use this convention. We further remark that the feature-wise anomaly scores $\alpha$ are related to *feature attribution scores* in explainability literature [33, 51, 55].

## 2.2 Formal Properties for Counterfactual Explanations

We now present the formal properties of counterfactual explanations. We will assume a given anomaly score function $s$ that is linearly decomposed by the feature-wise score function $\alpha$ and regularizer $\beta$. Given some input $x$, it is common to convert $\alpha(x) \in \mathbb{R}^n$ into a binary-valued vector $\omega(x) \in \{0, 1\}^n$ to classify which input feature is anomalous, and this is commonly done by a threshold:

$$\omega(x) = (\alpha_1(x) \geq \tau_1, \ldots, \alpha_n(x) \geq \tau_n), \quad \text{for some feature-wise threshold } \tau \in \mathbb{R}^n.$$

A binarization of the feature-wise scores suggests the need for region-specific anomaly scores, which we implement with the following indexing scheme on $s(x)$:

$$s_z(x) = \beta(x) + \sum_{i:z_i=1} \alpha_i(x), \quad \text{for all } z \in \{0, 1\}^n.$$

Then, the anomalous and non-anomalous regions have scores $s_{\omega(x)}(x)$ and $s_{\overline{\omega}(x)}(x)$, respectively, where $\overline{\omega}(x) = 1 - \omega(x)$ denotes non-anomalous region. We next enumerate some common desiderata of counterfactual explanations, where we will refer to the anomalous input as $x_{\mathsf{bad}}$ and the repaired version as $x_{\mathsf{fix}}$.

**Property 1 (Overall Improvement): The anomaly score should improve.** Because a "repaired" version should fix the anomaly by definition, one would reasonably expect that:

$$s(x_{\mathsf{fix}}) < s(x_{\mathsf{bad}}). \tag{P1}$$

**Property 2 (Similarity): The repair should resemble the original.** When $s$ is generated by a complex machine-learning model, it may be the case that it has a value $x_{\mathsf{fix}}$ where $s(x_{\mathsf{fix}}) \ll s(x_{\mathsf{bad}})$, but $x_{\mathsf{fix}}$ and $x_{\mathsf{bad}}$ bear little resemblance. In the case of vision models, $x_{\mathsf{fix}}$ may even resemble static noise. Such extreme dissimilarities between $x_{\mathsf{fix}}$ and $x_{\mathsf{bad}}$ are not desirable because a user cannot be expected to feasibly interpret this information. Thus, we desire a similarity condition as:

$$\overline{\omega}(x_{\mathsf{bad}}) \odot x_{\mathsf{bad}} \approx \overline{\omega}(x_{\mathsf{bad}}) \odot x_{\mathsf{fix}}, \tag{P2}$$

where $\odot$ denotes element-wise vector multiplication. This similarity condition states that the non-anomalous regions of the original and the repair, as given by $\omega(x_{\mathsf{bad}})$, should remain similar.

**Property 3 (Localized Improvement): The anomalous region should improve.** However, P1 and P2 are not sufficient. For example, one might have $s(x_{\mathsf{fix}}) < s(x_{\mathsf{bad}})$, but have a higher score on the anomalous region $\omega(x_{\mathsf{bad}})$. This is not desirable because it means that $x_{\mathsf{fix}}$ has not actually fixed the anomalous region of $x_{\mathsf{bad}}$. To ensure progress, we would like:

$$s_{\omega(x_{\mathsf{bad}})}(x_{\mathsf{fix}}) < s_{\omega(x_{\mathsf{bad}})}(x_{\mathsf{bad}}). \tag{P3}$$

**Property 4 (Non-degradation): The non-anomalous region should not significantly worsen.** Even when the above properties are satisfied, it is possible that the proposed repair could inadvertently increase the anomaly score on the non-anomalous region $\overline{\omega}(x_{\mathsf{bad}})$. This would mean repairing the anomalous region at the cost of corrupting the non-anomalous parts. We thus state the property against this as follows, where $\delta_4 > 0$ is a given tolerance threshold:

$$s_{\overline{\omega}(x_{\mathsf{bad}})}(x_{\mathsf{fix}}) \leq s_{\overline{\omega}(x_{\mathsf{bad}})}(x_{\mathsf{bad}}) + \delta_4. \tag{P4}$$

The benefit of our above formulation is that it encapsulates general, domain-independent desiderata of anomaly repairs. Importantly, this is achieved under the mild assumption of a linearly decomposable anomaly detector. We comment that some overlap among our proposed formal properties may arise in certain scenarios, and alternative sets could be more tailored for specific applications. Our goal, however, is to provide a foundational set of properties that ensures broad applicability, allowing further customization to suit individual application needs.

## 3 Property-guided Generation of Counterfactual Explanations

We now outline the process of conducting a property-guided repair for an anomalous input. In Section 3.1, we first introduce the generalized setup, where we define the four previously outlined properties as objective functions and frame the problem using risk-constrained optimization. Although this formulation clarifies the objectives, it is generally intractable. Therefore, in Section 3.2, we propose a diffusion-based algorithm to approximate the solution and achieve high-quality repairs.

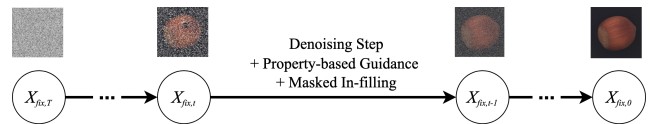

Figure 3: We run property-guided diffusion with masked in-filling.

## 3.1 A Generalized Formulation with Properties as Loss Functions

We first present a generalized setup for generating repairs. We conceptualize this in terms of a repair model $\mathcal{R}_{s,\omega}$ parametrized by an anomaly score function $s : \mathbb{R}^n \to \mathbb{R}$ with known linear decompositions $\alpha, \beta$ and a feature-wise binarization $\omega$. To generate a repair, we sample the model:

$$x_{\text{fix}} \sim \mathcal{R}_{s,\omega}(x_{\text{bad}}),$$

where a probabilistic formulation is relevant in the context of variational auto-encoders [27] or diffusion models [24]. However, it is important that $x_{\text{fix}}$ obeys the formal properties as outlined in Section 2.2. To do this, we cast these properties as loss functions listed below:

$$L_1 = s(x_{\text{fix}}) \tag{P1 loss}$$
$$L_2 = \|\overline{\omega}(x_{\text{bad}}) \odot (x_{\text{fix}} - x_{\text{bad}})\|_2 \tag{P2 loss}$$
$$L_3 = \max\{0, s_{\omega(x_{\text{bad}})}(x_{\text{fix}}) - s_{\omega(x_{\text{bad}})}(x_{\text{bad}})\} \tag{P3 loss}$$
$$L_4 = \max\{0, s_{\overline{\omega}(x_{\text{bad}})}(x_{\text{fix}}) - s_{\overline{\omega}(x_{\text{bad}})}(x_{\text{bad}}) - \delta_4\} \tag{P4 loss}$$

Our rationale is as follows. First, because the primary objective of anomaly repair is to reduce the anomaly score, we set $L_1$ as simply the score of $x_{\text{fix}}$. Second, we would like to ensure that $x_{\text{fix}}$ and $x_{\text{bad}}$ are similar in the non-anomalous region, and so formulate $L_2$ as the $\ell^2$ distance between $x_{\text{fix}}$ and $x_{\text{bad}}$ over $\overline{\omega}(x_{\text{bad}})$. Third, we formulate $L_3$ to apply a penalty when the anomalous region degrades in performance. Fourth, we allow for a degradation of score in the non-anomalous region $\overline{\omega}(x_{\text{bad}})$, up to some tolerance threshold $\delta_4$. We cast these as a risk-constrained optimization problem as follows:

$$\begin{aligned} \underset{\theta}{\text{minimize}} \quad & \underset{x_{\text{fix}} \sim \mathcal{R}_{s,\omega}(x_{\text{bad}};\theta)}{\mathbb{E}} s(x_{\text{fix}}) \\ \text{subject to} \quad & \underset{x_{\text{fix}} \sim \mathcal{R}_{s,\omega}(x_{\text{bad}};\theta)}{\mathbb{P}} \big[L_2 \le \delta_2, \ L_3 \le 0, \ L_4 \le 0\big] \ge 1 - \delta \end{aligned} \tag{1}$$

where $\theta$ is a parameter of the repair model, $\delta_2 > 0$ is a given threshold for $L_2$ and $\delta > 0$ is a given failure probability for the violation of at least one of (P2), (P3), or (P4). We acknowledge that multiple formulations for property-based losses are valid; however, the chosen approach is optimally suited to our context.

## 3.2 Formal Property-guided Diffusion

We adopt techniques from guided diffusion [19] to generate repairs $x_{\text{fix}}$. We first give a brief overview of a standard diffusion process and then adapt it to perform property-guided generation. We refer to [62] for a comprehensive guide on diffusion but attempt to make the exposition self-contained.

**Background.** A basic variant of diffusion models takes the form:

$$p(x_{t-1}|x_t) = \mathcal{N}(\mu_\theta(x_t, t), b_t^2 I), \quad \text{for } t = T, \dots, 1, \tag{2}$$

where $\mu_\theta$ is the *denoising model* with parameters $\theta$, and $b_1 < \cdots < b_T$ is the variance schedule. When $\mu_\theta$ is trained on non-anomalous data (see [62] for training details), one can generate non-anomalous samples of the training distribution by running the following iterative process:

$$x_T \sim \mathcal{N}(0, I), \quad x_{t-1} = \mu_\theta(x_t, t) + b_t z_t, \quad z_t \sim \mathcal{N}(0, I), \quad \text{for } t = T, \dots, 1, \tag{3}$$

where $x_T$ is the initial noise and $x_0$ is the output sample. The idea is to repeatedly remove noise from a Gaussian $x_T$ using $\mu_\theta$ until the final $x_0$ resembles a high-resolution image without defects, for instance. The iterations (3) is also known as *backward process*.

**Property-guided Diffusion.** We next show how to adapt the denoising iterations (3) to produce repairs. We use two main ideas: first, we use guidance [19] to slightly nudge the iterates $x_{t-1}$ of (3)

at every step using a property-based loss, to encourage that the final iterate $x_0$, which we take to be $x_{\text{fix}}$, is more amenable to our properties. Second, we used masked-infilling [39] to ensure that the non-anomalous region $\overline{\omega}(x_{\text{bad}})$ is generally preserved by the iterates. We implement this modified iteration as follows: beginning from the initial noise $x_{\text{fix},T} \sim \mathcal{N}(0, I)$, let:

$$\hat{x}_{\text{fix},t-1} = \underbrace{\mu_\theta(x_{\text{fix},t}, t) + b_t z_t}_{\text{Denoising step}} - \underbrace{\eta_t \nabla L(x_{\text{fix},t})}_{\text{Guidance term}}, \quad z_t \sim \mathcal{N}(0, I), \tag{4}$$

$$x_{\text{bad},t} = \sqrt{a_t} x_{\text{bad}} + \sqrt{1 - a_t} \epsilon_t, \quad \epsilon_t \sim \mathcal{N}(0, I), \tag{5}$$

$$x_{\text{fix},t-1} = \overline{\omega}(x_{\text{bad}}) \odot x_{\text{bad},t} + \omega(x_{\text{bad}}) \odot \hat{x}_{\text{fix},t-1}, \tag{6}$$

for $t = T, \ldots, 1$, where $a_t = \prod_{i=1}^t (1 - b_i)$ and $\eta_1 < \cdots < \eta_T$ is the guidance schedule. The property-based loss is given by:

$$L(x_{\text{fix},t}) = \lambda_1 L_1 + \lambda_2 L_2 + \lambda_3 L_3 + \lambda_4 L_4, \tag{7}$$

with $L_1, L_2, L_3, L_4$ as in Section 3.1 and weights $\lambda_1, \lambda_2, \lambda_3, \lambda_4 > 0$. In the above, (4) first generates $\hat{x}_{\text{fix},t-1}$ from $x_{\text{fix},t}$ by combining a standard denoising step with a guidance term. Then, (6) combines $\hat{x}_{\text{fix},t-1}$ with $\hat{x}_{\text{bad},t}$, from (5), in a masked-infilling operation [39] to yield $x_{\text{fix},t-1}$. This masked in-filling ensures similarity between $x_{\text{bad}}$ and $x_{\text{fix}}$ (i.e., $x_{\text{fix},0}$) over the non-anomalous region $\overline{\omega}(x_{\text{bad}})$.

We emphasize that the diffusion iterations given by (4), (5), (6), and do not guarantee the satisfaction of our formal properties. Rather, these iterations tend toward an output that better respects these properties — as we later shown in our experiments. In particular, our property-based losses define a way to evaluate the quality to which each property is satisfied or violated.

## 4 Experiments

Our experiments evaluate the performance of AR-Pro across vision and time-series datasets. In particular, we aim to address the following research questions:

- **(RQ1) Empirical Validation**: How well does AR-Pro repair anomalies for different domains? In particular, we investigate how well the four properties are satisfied when the diffusion process is guided or unguided, as in the baseline.

- **(RQ2) Ablation Study**: How do the different hyper-parameters affect the repair quality? We focus on the weights $\lambda_1, \lambda_2, \lambda_3, \lambda_4$ for the four property-based losses.

**Vision Anomaly Models and Datasets.** For anomaly detectors, we used the anomalib [4] implementation of Fastflow [66] (with ResNet-50-2 backbone [68]) and Efficient-AD [11]. For datasets, we used the VisA [73] and MVTec-AD [13] datasets. VisA and MVTec-AD involve anomaly detection in the context of industrial manufacturing, where VisA consists of 12 image classes, and MVTec-AD consists of 15 image classes. Both FastFlow and Efficient-AD were trained with AdamW and a learning rate of $10^{-4}$ until convergence.

**Time-series Anomaly Datasets and Models.** For anomaly detectors, we used the GPT-2 [48] and Llama2 [59] architectures for time-series anomaly detection. In particular, we use only the first 6 layers of GPT-2 and the first 4 layers of Llama-2 (with an embedding dimension of 1024) to accelerate training. For datasets, we used the SWaT (51 features) [38], HAI (86 features) [54], and WADI (127 features) [2] datasets, split into sliding windows of size 100. Both our versions of GPT-2 and Llama-2 were trained with AdamW and a learning rate of $10^{-5}$ until convergence.

**Diffusion-based Repair Models.** We used the HuggingFace implementation of DDPM [24] for vision data and Diffusion-TS [67] for time-series data. Both models were trained on the non-anomalous instances of their respective datasets using AdamW and a learning rate of $10^{-4}$ until convergence.

**Evaluation Metrics.** We use the four property-based loss functions defined in Section 3.1 as our evaluation metrics. In particular, we measure the improvement of property-guided diffusion over un-guided diffusion. We adapt these metrics below, where we write $\omega$ to mean $\omega(x_{\text{bad}})$ for brevity:

- **Property 1 (Overall Improvement):** $M_s \equiv s(x_{\text{fix}})$.
- **Property 2 (Similarity):** $M_d \equiv \|\overline{\omega} \odot (x_{\text{fix}} - x_{\text{bad}})\|_2$
- **Property 3 (Localized Improvement):** $M_\omega \equiv s_\omega(x_{\text{fix}}) - s_\omega(x_{\text{bad}})$
- **Property 4 (Non-degradation):** $M_{\overline{\omega}} \equiv s_{\overline{\omega}}(x_{\text{fix}}) - s_{\overline{\omega}}(x_{\text{bad}})$

| Dataset | Class | $M_s(\downarrow)$ | | $M_d(\downarrow)$ | | $M_\omega(\downarrow)$ | | $M_{\varpi}(\downarrow)$ | |
|---|---|---|---|---|---|---|---|---|---|
| | | Baseline | Guided | Baseline | Guided | Baseline | Guided | Baseline | Guided |
| **VisA** | candle | -23.39 ± 1.23 | -26.43 ± 0.14 | 336.06 ± 44.41 | 8.34 ± 0.06 | -0.003 ± 0.002 | -0.007 ± 0.001 | 0.05 ± 0.01 | -0.02 ± 0.004 |
| | capsules | 12.45 ± 30.02 | -17.86 ± 0.20 | 426.69 ± 45.30 | 8.41 ± 0.01 | 0.009 ± 0.003 | -0.005 ± 0.001 | 0.15 ± 0.02 | -0.01 ± 0.002 |
| | cashew | -16.08 ± 0.85 | -17.13 ± 0.33 | 307.39 ± 63.72 | 8.40 ± 0.01 | -0.002 ± 0.004 | -0.005 ± 0.001 | 0.06 ± 0.03 | -0.00 ± 0.002 |
| | chewinggum | -14.56 ± 0.89 | -15.70 ± 0.14 | 258.32 ± 53.66 | 8.41 ± 0.01 | -0.007 ± 0.003 | -0.009 ± 0.003 | 0.02 ± 0.02 | -0.01 ± 0.002 |
| | fryum | -13.54 ± 3.02 | -18.93 ± 0.42 | 349.18 ± 32.39 | 8.36 ± 0.01 | 0.003 ± 0.005 | -0.007 ± 0.001 | 0.12 ± 0.04 | -0.01 ± 0.003 |
| | macaroni1 | 199.95 ± 145.07 | -25.63 ± 0.25 | 658.18 ± 81.65 | 8.41 ± 0.01 | 0.012 ± 0.002 | -0.005 ± 0.001 | 0.18 ± 0.02 | -0.01 ± 0.004 |
| | macaroni2 | -15.61 ± 11.75 | -25.18 ± 0.17 | 550.91 ± 55.19 | 8.41 ± 0.01 | 0.004 ± 0.003 | -0.004 ± 0.001 | 0.11 ± 0.02 | -0.01 ± 0.002 |
| | pcb1 | -23.49 ± 0.49 | -25.04 ± 1.08 | 325.54 ± 50.82 | 8.38 ± 0.01 | -0.005 ± 0.006 | -0.006 ± 0.003 | 0.05 ± 0.02 | -0.01 ± 0.004 |
| | pcb2 | -18.20 ± 0.67 | -19.15 ± 0.12 | 289.54 ± 48.32 | 8.41 ± 0.01 | -0.005 ± 0.003 | -0.005 ± 0.001 | 0.03 ± 0.02 | -0.01 ± 0.002 |
| | pcb3 | -19.87 ± 4.29 | -24.19 ± 0.15 | 291.16 ± 50.72 | 8.41 ± 0.01 | -0.003 ± 0.004 | -0.005 ± 0.002 | 0.05 ± 0.04 | -0.01 ± 0.002 |
| | pcb4 | 2.51 ± 23.68 | -20.24 ± 0.09 | 337.12 ± 75.48 | 8.41 ± 0.01 | 0.000 ± 0.004 | -0.005 ± 0.001 | 0.12 ± 0.05 | -0.01 ± 0.003 |
| | pipefryum | -15.08 ± 3.83 | -19.80 ± 0.22 | 252.68 ± 78.99 | 8.41 ± 0.01 | -0.002 ± 0.006 | -0.008 ± 0.001 | 0.09 ± 0.05 | -0.02 ± 0.005 |
| | Δ(↑) | **+26.54%** | | **+97.46%** | | **+146.42%** | | **+114.16%** | |
| **MVTec-AD** | bottle | -13.67 ± 0.03 | -13.16 ± 1.42 | 252.11 ± 41.14 | 8.70 ± 1.76 | -0.016 ± 0.001 | -0.012 ± 0.001 | -0.05 ± 0.01 | -0.05 ± 0.003 |
| | cable | -12.56 ± 0.13 | -13.18 ± 0.17 | 206.08 ± 80.64 | 5.03 ± 1.94 | -0.006 ± 0.003 | -0.007 ± 0.003 | 0.02 ± 0.01 | -0.01 ± 0.005 |
| | capsule | -13.73 ± 0.35 | -15.21 ± 0.31 | 105.41 ± 48.80 | 5.07 ± 2.23 | -0.004 ± 0.003 | -0.006 ± 0.002 | 0.04 ± 0.01 | -0.01 ± 0.006 |
| | carpet | -18.93 ± 0.35 | -19.47 ± 0.70 | 201.02 ± 82.18 | 4.98 ± 2.04 | -0.014 ± 0.005 | -0.014 ± 0.005 | 0.00 ± 0.02 | -0.01 ± 0.010 |
| | grid | 40.78 ± 115.98 | -13.73 ± 0.11 | 556.52 ± 65.32 | 10.01 ± 0.85 | 0.010 ± 0.002 | -0.010 ± 0.002 | 0.17 ± 0.02 | -0.02 ± 0.004 |
| | hazelnut | -10.34 ± 0.23 | -12.94 ± 0.21 | 127.33 ± 52.05 | 5.07 ± 1.98 | -0.003 ± 0.004 | -0.011 ± 0.004 | 0.09 ± 0.01 | -0.02 ± 0.005 |
| | leather | -12.81 ± 0.14 | -13.85 ± 0.08 | 484.22 ± 75.46 | 10.01 ± 1.30 | -0.013 ± 0.002 | -0.014 ± 0.002 | 0.02 ± 0.01 | -0.02 ± 0.005 |
| | metal nut | -8.94 ± 0.34 | -11.05 ± 0.33 | 214.22 ± 104.32 | 5.71 ± 2.74 | -0.001 ± 0.004 | -0.007 ± 0.002 | 0.07 ± 0.02 | -0.02 ± 0.003 |
| | pill | -10.90 ± 0.08 | -11.81 ± 0.17 | 242.05 ± 20.04 | 10.88 ± 0.79 | -0.006 ± 0.001 | -0.007 ± 0.001 | 0.01 ± 0.01 | -0.02 ± 0.001 |
| | screw | -11.28 ± 0.28 | -13.10 ± 0.07 | 432.16 ± 66.10 | 10.16 ± 1.49 | -0.001 ± 0.001 | -0.006 ± 0.001 | 0.05 ± 0.01 | -0.02 ± 0.002 |
| | tile | -11.25 ± 0.44 | -12.18 ± 0.57 | 425.23 ± 44.84 | 9.91 ± 1.03 | -0.012 ± 0.002 | -0.015 ± 0.004 | 0.04 ± 0.01 | -0.03 ± 0.006 |
| | toothbrush | -6.43 ± 0.10 | -8.33 ± 0.05 | 151.76 ± 25.94 | 9.27 ± 1.77 | -0.002 ± 0.002 | -0.009 ± 0.002 | 0.05 ± 0.00 | -0.03 ± 0.003 |
| | transistor | -12.15 ± 0.18 | -12.64 ± 5.51 | 278.70 ± 69.53 | 7.27 ± 1.83 | -0.007 ± 0.002 | -0.007 ± 0.001 | 0.02 ± 0.01 | -0.01 ± 0.002 |
| | wood | -9.65 ± 2.29 | -13.54 ± 0.26 | 344.91 ± 67.58 | 10.17 ± 1.75 | -0.013 ± 0.003 | -0.019 ± 0.001 | 0.03 ± 0.02 | -0.02 ± 0.005 |
| | zipper | -12.62 ± 0.19 | -13.24 ± 0.05 | 314.25 ± 34.11 | 10.10 ± 1.27 | -0.009 ± 0.001 | -0.009 ± 0.001 | -0.00 ± 0.00 | -0.03 ± 0.002 |
| | Δ(↑) | **+8.35%** | | **+97.34%** | | **+29.15%** | | **+154.76%** | |

Table 1: AR-Pro outperforms a non-guided diffusion baseline across our four metrics. We show the results for all VisA and MVTec-AD categories, where $\Delta$ is the median percentage improvement.

Each experiment employs a representative anomaly detector and dataset with predefined train-test splits. The performance of the anomaly detectors is detailed in Appendix B. For $\omega$, each feature-wise threshold $\tau_i$ is taken to be the 90% quantile of the training set's feature-wise anomaly scores.

## 4.1 (RQ1) Empirical Validation

We now evaluate the performance of AR-Pro for generating anomaly repairs with respect to the four evaluation metrics. We report the mean and standard deviation values. Because there may be a large range of values across classes, we report the median improvement ($\Delta$) of the guided generation over the baseline. We present results for the FastFlow anomaly detector on the VisA and MVTec dataset in Table 1, and refer to Appendix C for additional results with Efficient-AD.

**Quantitative Results.** For all the categories, the guided results show significant improvement over the baseline on the formal criteria, with an average improvement of 84.27%. There is a wide range of $M_s$ values for baseline models in certain classes, possibly due to deviations in the color scheme of the generated images from their training distribution. Hence, we report the median improvement across the classes in the last row to mitigate the impact of outliers.

We show results for time-series data in Table 2. AR-Pro achieves lower error on $M_d$, $M_\omega$, and $M_{\overline{\omega}}$, with an overall improvement of 60.03% over the baseline. Specifically, Llama-2 achieves an average improvement of 67.17% in formal metrics, while GPT-2 increases by 53.90% on average. Compared to image data, the performance on $M_d$ is not as competitive and exhibits considerable variability, likely due to the broader range of adjustments required for repairs to ensure smooth signals.

In addition, we evaluate whether the guided repairs generate non-anomalous samples by comparing them against a conformity threshold with 95% confidence derived from the training set [9]. Treating the non-anomalous class as the negative, we report the True Negative Rate (TNR) in Table 3. The TNR of most VisA and MVTec-AD categories reaches 100%, surpassing the majority of baseline values. As a result, the guided repair achieves a median TNR of 100% for both VisA and MVTec, representing an average improvement of 2.50% over the baseline. Across the three time-series datasets (SWaT, HAI, and WADI), the guided repair obtains an average TNR of 95.33%, which is 92.66% higher than the baseline average of 2.67%. Overall, 99.26% of guided repairs across both domains are classified as non-anomalous with 95% confidence, statistically confirming the effectiveness of AR-Pro.

**Qualitative Results.** We present qualitative examples here to illustrate that our method can generate semantically meaningful repairs, as shown in Figure 4 for VisA and Figure 5 for MVTec-AD. For

| Model | Dataset | $M_s(\downarrow)$ | | $M_d(\downarrow)$ | | $M_\omega(\downarrow)$ | | $M_\varpi(\downarrow)$ | |
|---|---|---|---|---|---|---|---|---|---|
| | | Baseline | Guided | Baseline | Guided | Baseline | Guided | Baseline | Guided |
| **Llama2** | SWaT | $0.83 \pm 0.05$ | $0.59 \pm 0.04$ | $3.05 \pm 0.61$ | $7.25 \pm 2.50$ | $0.084 \pm 0.026$ | $-0.026 \pm 0.008$ | $0.19 \pm 0.02$ | $0.05 \pm 0.012$ |
| | WADI | $0.97 \pm 0.00$ | $0.26 \pm 0.01$ | $0.98 \pm 0.01$ | $0.00 \pm 0.00$ | $0.087 \pm 0.005$ | $0.000 \pm 0.000$ | $0.61 \pm 0.01$ | $0.00 \pm 0.000$ |
| | HAI | $0.92 \pm 0.00$ | $0.58 \pm 0.01$ | $0.75 \pm 0.00$ | $0.00 \pm 0.00$ | $0.166 \pm 0.008$ | $0.000 \pm 0.000$ | $0.18 \pm 0.01$ | $0.00 \pm 0.000$ |
| | $\Delta(\uparrow)$ | **+46.36%** | | **+20.77%** | | **+110.32%** | | **+91.23%** | |
| **GPT-2** | SWaT | $0.68 \pm 0.06$ | $0.57 \pm 0.05$ | $12.81 \pm 15.82$ | $16.28 \pm 16.30$ | $-0.029 \pm 0.041$ | $-0.094 \pm 0.038$ | $0.13 \pm 0.04$ | $0.09 \pm 0.031$ |
| | WADI | $0.71 \pm 0.04$ | $0.32 \pm 0.00$ | $34.45 \pm 0.60$ | $42.15 \pm 0.94$ | $0.018 \pm 0.003$ | $-0.019 \pm 0.002$ | $0.43 \pm 0.04$ | $0.07 \pm 0.002$ |
| | HAI | $0.92 \pm 0.02$ | $0.61 \pm 0.11$ | $3.69 \pm 9.72$ | $5.67 \pm 18.80$ | $0.136 \pm 0.050$ | $-0.005 \pm 0.018$ | $0.21 \pm 0.03$ | $0.04 \pm 0.127$ |
| | $\Delta(\uparrow)$ | **+34.93%** | | **-34.37%** | | **+177.79%** | | **+37.24%** | |

Table 2: Comparison of baseline and guided performance across four metrics for SWaT, WADI, and HAI dataset categories with Llama2 and GPT2 model. $\Delta$ is the median improvement percentage of the guided result from baseline.

| Dataset | Category | Baseline TNR ($\uparrow$) | Guided TNR ($\uparrow$) | Category | Baseline TNR ($\uparrow$) | Guided TNR ($\uparrow$) |
|---|---|---|---|---|---|---|
| VisA | Candle | **1.00** | **1.00** | Fryum | 0.66 | **1.00** |
| | Capsules | 0.00 | **1.00** | Pipe Fryum | **1.00** | **1.00** |
| | Cashew | **1.00** | **1.00** | PCB 1 | **1.00** | 0.96 |
| | Chewinggum | **1.00** | **1.00** | PCB 2 | 1.00 | **1.00** |
| | Macaroni 1 | 0.11 | **1.00** | PCB 3 | 0.90 | **1.00** |
| | Macaroni 2 | 0.52 | **1.00** | PCB 4 | 0.18 | **1.00** |
| MVTec-AD | Bottle | **1.00** | **1.00** | Grid | 0.00 | **1.00** |
| | Cable | **1.00** | **1.00** | Hazelnut | **1.00** | **1.00** |
| | Capsule | **1.00** | **1.00** | Leather | **1.00** | **1.00** |
| | Carpet | **1.00** | **1.00** | Metal Nut | **1.00** | **1.00** |
| | Pill | **1.00** | **1.00** | Screw | **1.00** | **1.00** |
| | Tile | **1.00** | **1.00** | Toothbrush | **1.00** | 0.96 |
| | Transistor | 0.56 | **1.00** | Wood | **1.00** | **1.00** |
| | Zipper | **1.00** | **1.00** | | | |
| Overvall Median | VisA | 0.95 | **1.00** | MVTec-AD | **1.00** | **1.00** |
| | SWaT | 0.08 | **0.86** | HAI | 0.00 | **1.00** |
| | WADI | 0.00 | **1.00** | | | |

Table 3: True Negative Rate (TNR) for categories in the vision (VisA, MVTec) and time-series (SWaT, HAI, and WADI) anomaly detection datasets.

example in Figure 4, we observe that for categories PCB 1, PCB 2, PCB 3, and PCB 4, the baseline fails to rigorously repair the anomaly: the orientation of the board is reversed for PCB 1; unintended white marks appear on the lower left of PCB 2; the dark lines in the potentiometer change for PCB 3; and the "FC-75" label is missing for PCB 4. However, by integrating formal property guidance, our approach accurately reconstructs all these details while effectively removing the anomalies. For the MVTec-AD examples, AR-Pro produces repairs that more closely resemble the original inputs compared to those generated by the baseline. This demonstrates that our method's generated repairs adhere more rigorously to the formal properties.

In Figure 6, we present examples from our time-series repair, where the anomalous time segment is shaded in red. Our results demonstrate that AR-Pro generates a signal that resembles the original signal better, as shown in the first two plots of Figure 6. In addition, we recover the sensor time series to non-anomalous values when the baseline fails to repair the anomaly, as shown in the last two plots of Figure 6.

More repair examples are available in Appendix D. However, although the quality of the generated repairs has improved, we notice that this enhancement comes with a trade-off of increased inference time. Further details can be found in Appendix E.

## 4.2 (RQ2) Ablation Study

We randomly sampled 100 instances to compute the mean of each metric in order to evaluate the effect of hyper-parameters $\lambda_1, \lambda_2, \lambda_3, \lambda_4$ associated with each property-based loss. Each line in the plots represents results obtained while keeping the other hyper-parameters at 1.0. The ablation results for the time series are presented in Figure 7, with additional plots in Appendix F.

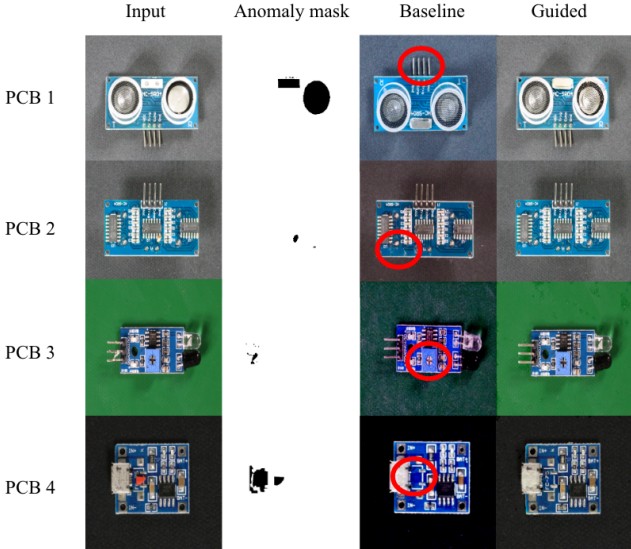

Figure 4: The original input and ground truth anomaly mask are displayed in the first two columns. The baseline method fails to preserve close similarity to the input PCB boards, as highlighted in the third column. Guided vision repair examples in the fourth column address these deficiencies.

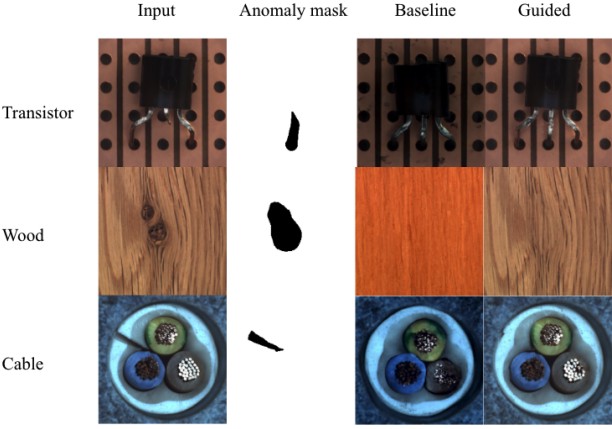

Figure 5: MVTec repairs with AR-Pro; better resemble the original compared to the baseline.

Our observations indicate that variations in the scale of property hyperparameters do not significantly impact the formal metrics, as the range of change remains relatively small. In addition, no consistent trends were observed when varying the $\lambda_1, \lambda_2, \lambda_3, \lambda_4$ hyper-parameters. This suggests that our framework demonstrates robust performance, and extensive tuning may be unnecessary.

## 5 Related Work

Numerous techniques have been developed for anomaly detection across various domains [3, 49]. Traditional approaches to anomaly detection include clustering [43, 57] and statistical methods [34] such as ARIMA [41] and Gaussian models [52]. However, these methods often struggle with high-dimensional data. More recently, deep learning-based anomaly detection [20, 28], including autoencoders [27] and GANs [18], can detect high-dimensional anomalies via reconstruction error. For time-series data, LSTMs [32] and transformer-based models [60, 63, 69, 72] have shown exceptional performance. Additionally, diffusion models are emerging as promising tools for visual anomaly detection [42, 71]. While these methods vary in strengths and are continually improving, explaining

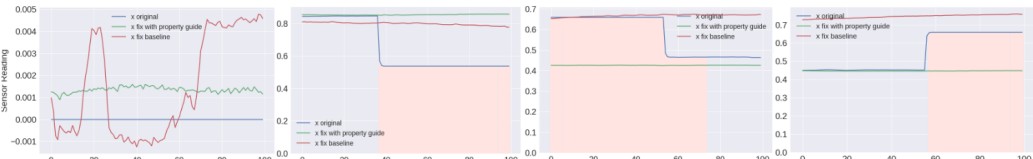

Figure 6: Original input is the blue line, property guided fix is the green line, and the baseline is the red line. The first image shows that the baseline generates a spurious signal when there is no anomaly. The second image shows that the baseline repairs the anomaly, but not as effectively as with guidance. The last two images show instances where the baseline fails to repair the anomaly.

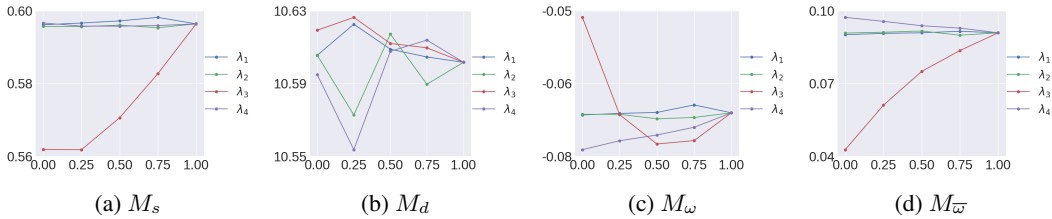

| (a) $M_s$ | (b) $M_d$ | (c) $M_\omega$ | (d) $M_{\overline{\omega}}$ |

Figure 7: Varying the hyper-parameters does significantly change $M_s$, $M_d$, $M_\omega$, and $M_{\overline{\omega}}$.

anomalies remains a challenge [10]. Most current methods rely on feature importance scores or visualizations [44], such as gradients or reconstructions [46], which often fail to provide actionable insights. The lack of formal frameworks [65] and consistent evaluation metrics [1] complicates this issue. For example, the absence of formal metrics leads to inconsistencies in evaluation [26], underscoring the need for more rigorous approaches and reliable criteria [37]. Most similar to our work is [56], which also performs time series-specific generation of counterfactual explanations in the form of anomaly repairs but considers a different set of properties and does not use diffusion. Our work also focuses on generative modeling to produce counterfactual explanations in the form of anomaly repairs. For vision, the current leading paradigms are diffusion models [24] and generative adversarial networks [23]. Diffusion models are also applicable to time-series data [67], and we refer to [22] for a survey on other techniques.

## 6    Discussion

The main theoretical contribution of our work is the identification of common counterfactual explanation desiderata for linearly decomposable anomaly detectors. While we have identified four formal properties, we acknowledge that other valid ones may also exist. Moreover, we recommend that practitioners evaluate and choose the properties necessary for the particular problem, and this is made possible by the form of our diffusion guidance function in (7). The quality of anomaly repairs depends on the performance of the anomaly detector and the generative model. While there may have been limitations in our efforts, we found it challenging to use variational auto-encoders [27] for generating high-quality repairs. Furthermore, our implementation is focused on diffusion models, but the ideas presented can also be extended to other generative techniques.

## 7    Conclusion

We present AR-Pro, a framework for generating and evaluating counterfactual explanations in anomaly detection. We use the fact that common anomaly detectors are linearly decomposable, which lets us define formal, general, domain-independent properties for explainability. Using these properties, we show how to generate high-quality counterfactuals using a property-guided diffusion setup. We demonstrate the effectiveness of AR-Pro on vision and time-series datasets and showcase our improvement over off-the-shelf diffusion models.

**Acknowledgement**    This work was supported in part by ARO MURI W911NF2010080, NSF-2125561, and NSF-2143274.

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

## A  Computational Resources

All experiments were done on a server with three NVIDIA GeForce RTX 4090 GPUs.

## B  Anomaly Detector Performance

In this section, we report the performance of anomaly detectors. The results for FastFlow [66] are presented in Table 4, and the results for GPT-2 [48] are shown in Table 5. We did not perform extensive parameter tuning, as the performance of anomaly detectors is not the primary focus of our work. In addition, it is recommended that users adhere to usage guidelines of using GPT-2 or implementing safety filters.

## C  Additional Experiments

We include additional model EfficientAD [11] and dataset MVTec [13]. The results are shown in Table 6. EfficientAD achieved an 88.24% average improvement across the four metrics on MVTec-AD and 74.85% improvement on VisA.

## D  More Qualitative Examples

Additional VisA examples can be found in Figure 8, while additional MVTec-AD examples can be found in Figure 9 and Figure 10. Additional time-series examples can be found in Figure 11.

## E  Inference Time

We randomly sampled 50 instances from the testing set to compare inference times, as shown in Table 7. We observed that although property guidance generates higher-quality repairs, it results in slightly longer inference times for time series data and significantly longer times for image data, possibly due to the higher dimensionality of the inputs. In total, it took about 40 hours to finish RQ1 for VisA and 25 hours to finish SWaT. For RQ2, it took 5 hours for SWaT. Improving computation time will be a focus of our future work.

## F  More Ablation Plots

In addition to $\lambda_1$ to $\lambda_4$, we also perform ablation on the overall guidance scale, which we denote using $\lambda_\phi$. The The ablation study for $\lambda_\phi$ on time series can be found in Figure 12. The ablation for image data, using cashew class as an example can be found in Figure 13 and Figure 14.

| Category | Image AUROC | Pixel AUROC |
|---|---|---|
| Candle | 0.6501 | 0.5617 |
| Capsules | 0.5352 | 0.8188 |
| Cashew | 0.3816 | 0.7558 |
| Chewing Gum | 0.3754 | 0.9362 |
| Fryum | 0.761 | 0.4244 |
| Macaroni1 | 0.2607 | 0.9127 |
| Macaroni2 | 0.504 | 0.7729 |
| PCB1 | 0.7173 | 0.9163 |
| PCB2 | 0.7398 | 0.7945 |
| PCB3 | 0.5361 | 0.8073 |
| PCB4 | 0.6331 | 0.6183 |
| Pipe Fryum | 0.4696 | 0.2799 |
| Average | 0.5470 | 0.7166 |

Table 4: AUROC Scores of Fastflow for Various Categories in VisA

| Metric | GPT-2 on SWaT |
|---|---|
| Accuracy | 0.9784 |
| Precision | 0.8831 |
| Recall | 0.9472 |
| F1-score | 0.9140 |

Table 5: Performance of GPT-2 anomaly detectors on SWaT Datasets

| Model | Class | $M_s(\downarrow)$ | | $M_d(\downarrow)$ | | $M_\omega(\downarrow)$ | | $M_{\overline{\omega}}(\downarrow)$ | |
|---|---|---|---|---|---|---|---|---|---|
| | | Baseline | Guided | Baseline | Guided | Baseline | Guided | Baseline | Guided |
| **EfficientAD** (MVTec-AD) | bottle | $24.91 \pm 1.73$ | $15.34 \pm 2.49$ | $252.67 \pm 25.10$ | $9.00 \pm 1.11$ | $0.014 \pm 0.462$ | $-0.388 \pm 0.411$ | $1.53 \pm 0.21$ | $-0.02 \pm 0.030$ |
| | cable | $6.35 \pm 1.00$ | $6.35 \pm 0.63$ | $377.28 \pm 55.71$ | $9.14 \pm 1.20$ | $-0.120 \pm 0.052$ | $-0.138 \pm 0.031$ | $0.32 \pm 0.07$ | $-0.03 \pm 0.009$ |
| | capsule | $141.59 \pm 3.43$ | $142.81 \pm 2.38$ | $223.74 \pm 26.83$ | $10.58 \pm 1.23$ | $-0.366 \pm 0.395$ | $-0.120 \pm 0.202$ | $-0.29 \pm 0.31$ | $-0.07 \pm 0.081$ |
| | carpet | $6.71 \pm 3.11$ | $0.70 \pm 0.04$ | $412.91 \pm 50.80$ | $10.17 \pm 1.28$ | $0.340 \pm 0.220$ | $-0.011 \pm 0.004$ | $2.23 \pm 1.34$ | $-0.00 \pm 0.001$ |
| | grid | $3.24 \pm 1.06$ | $0.32 \pm 0.02$ | $555.47 \pm 74.34$ | $10.03 \pm 0.59$ | $0.220 \pm 0.086$ | $-0.009 \pm 0.003$ | $1.56 \pm 0.58$ | $-0.00 \pm 0.001$ |
| | hazelnut | $94.40 \pm 7.79$ | $48.76 \pm 18.15$ | $238.92 \pm 26.92$ | $9.54 \pm 0.94$ | $2.380 \pm 1.002$ | $-0.856 \pm 0.775$ | $20.99 \pm 1.98$ | $0.05 \pm 0.061$ |
| | leather | $37.15 \pm 2.45$ | $16.42 \pm 0.63$ | $481.82 \pm 68.25$ | $9.99 \pm 1.43$ | $1.347 \pm 0.373$ | $-0.249 \pm 0.186$ | $12.03 \pm 1.86$ | $0.01 \pm 0.014$ |
| | metal nut | $3.51 \pm 0.09$ | $2.56 \pm 0.11$ | $421.94 \pm 56.50$ | $10.40 \pm 1.32$ | $0.024 \pm 0.033$ | $-0.025 \pm 0.035$ | $0.63 \pm 0.05$ | $-0.00 \pm 0.002$ |
| | pill | $4.56 \pm 0.09$ | $2.38 \pm 0.11$ | $261.21 \pm 15.65$ | $10.88 \pm 0.70$ | $0.178 \pm 0.026$ | $-0.014 \pm 0.020$ | $0.55 \pm 0.04$ | $-0.01 \pm 0.005$ |
| | screw | $1.05 \pm 0.20$ | $0.98 \pm 0.34$ | $376.92 \pm 40.00$ | $10.22 \pm 1.00$ | $0.020 \pm 0.004$ | $0.006 \pm 0.004$ | $0.10 \pm 0.00$ | $0.01 \pm 0.001$ |
| | tile | $6.67 \pm 1.98$ | $0.82 \pm 0.27$ | $432.18 \pm 62.29$ | $9.85 \pm 1.47$ | $0.094 \pm 0.308$ | $-0.318 \pm 0.236$ | $2.75 \pm 1.08$ | $-0.02 \pm 0.012$ |
| | toothbrush | $50.02 \pm 5.44$ | $35.26 \pm 9.41$ | $142.70 \pm 25.12$ | $9.27 \pm 1.69$ | $-0.175 \pm 0.265$ | $-0.013 \pm 0.056$ | $1.13 \pm 0.63$ | $-0.02 \pm 0.011$ |
| | transistor | $37.40 \pm 3.66$ | $46.66 \pm 10.74$ | $270.86 \pm 72.84$ | $26.60 \pm 100.98$ | $-1.355 \pm 0.672$ | $-0.311 \pm 0.485$ | $-1.78 \pm 1.96$ | $-0.35 \pm 1.506$ |
| | wood | $91.43 \pm 54.07$ | $22.96 \pm 2.72$ | $353.12 \pm 47.96$ | $10.24 \pm 1.24$ | $1.829 \pm 0.560$ | $-0.489 \pm 0.487$ | $14.30 \pm 3.73$ | $-0.15 \pm 0.066$ |
| | zipper | $1.37 \pm 0.20$ | $0.52 \pm 0.12$ | $305.86 \pm 32.46$ | $9.91 \pm 1.01$ | $0.030 \pm 0.009$ | $-0.015 \pm 0.003$ | $0.25 \pm 0.02$ | $-0.00 \pm 0.002$ |
| | $\Delta(\uparrow)$ | **+47.85%** | | **+97.10%** | | **+107.84%** | | **+100.16%** | |
| **EfficientAD** (VisA) | candle | $280.37 \pm 48.14$ | $387.79 \pm 60.55$ | $166.18 \pm 27.27$ | $8.71 \pm 1.27$ | $-7.219 \pm 3.677$ | $-0.212 \pm 0.379$ | $-6.98 \pm 7.23$ | $-0.22 \pm 0.285$ |
| | capsules | $28.98 \pm 10.27$ | $31.01 \pm 5.50$ | $371.05 \pm 52.72$ | $8.41 \pm 0.01$ | $-0.570 \pm 0.227$ | $-0.036 \pm 0.042$ | $2.06 \pm 0.74$ | $-0.01 \pm 0.012$ |
| | cashew | $364.66 \pm 35.11$ | $363.13 \pm 31.62$ | $255.69 \pm 54.96$ | $8.76 \pm 1.85$ | $-2.968 \pm 4.098$ | $-2.778 \pm 3.437$ | $-0.32 \pm 2.53$ | $0.06 \pm 1.063$ |
| | chewinggum | $81.47 \pm 4.01$ | $75.87 \pm 11.40$ | $179.61 \pm 15.37$ | $8.40 \pm 0.01$ | $-0.216 \pm 0.498$ | $-0.176 \pm 0.486$ | $0.76 \pm 0.53$ | $-0.02 \pm 0.020$ |
| | fryum | $267.90 \pm 55.84$ | $130.03 \pm 18.19$ | $222.72 \pm 32.44$ | $8.38 \pm 0.01$ | $5.766 \pm 2.300$ | $-0.539 \pm 1.032$ | $9.62 \pm 2.04$ | $-0.08 \pm 0.348$ |
| | macaroni1 | $881.24 \pm 32.80$ | $657.44 \pm 16.24$ | $227.79 \pm 13.91$ | $8.41 \pm 0.01$ | $16.696 \pm 3.118$ | $-1.117 \pm 0.735$ | $25.61 \pm 4.36$ | $-0.18 \pm 0.195$ |
| | macaroni2 | $350.04 \pm 63.90$ | $198.60 \pm 29.18$ | $244.22 \pm 25.98$ | $8.41 \pm 0.01$ | $3.070 \pm 2.147$ | $-0.523 \pm 0.799$ | $3.65 \pm 1.63$ | $0.17 \pm 0.375$ |
| | pcb1 | $24.09 \pm 2.89$ | $15.87 \pm 0.56$ | $208.50 \pm 31.13$ | $8.37 \pm 0.02$ | $0.293 \pm 0.068$ | $-0.001 \pm 0.027$ | $0.21 \pm 0.09$ | $-0.00 \pm 0.003$ |
| | pcb2 | $10.61 \pm 0.67$ | $9.07 \pm 1.22$ | $496.76 \pm 52.56$ | $8.41 \pm 0.01$ | $0.125 \pm 0.157$ | $-0.040 \pm 0.029$ | $3.66 \pm 0.38$ | $-0.01 \pm 0.005$ |
| | pcb3 | $15.06 \pm 8.65$ | $27.82 \pm 3.88$ | $363.07 \pm 38.92$ | $8.41 \pm 0.01$ | $-0.708 \pm 0.368$ | $-0.007 \pm 0.011$ | $-1.34 \pm 0.95$ | $-0.01 \pm 0.007$ |
| | pcb4 | $61.88 \pm 14.91$ | $79.72 \pm 20.28$ | $298.36 \pm 27.41$ | $8.41 \pm 0.01$ | $0.020 \pm 0.204$ | $-0.110 \pm 0.179$ | $1.53 \pm 0.19$ | $0.02 \pm 0.014$ |
| | pipe fryum | $112.89 \pm 16.76$ | $111.60 \pm 8.36$ | $117.01 \pm 33.67$ | $8.41 \pm 0.01$ | $0.619 \pm 1.512$ | $0.018 \pm 1.140$ | $0.40 \pm 0.97$ | $-0.20 \pm 0.289$ |
| | $\Delta(\uparrow)$ | **+4.01%** | | **+96.43%** | | **+98.65%** | | **+100.30%** | |

Table 6: Comparison of baseline and guided performance across four metrics for MVTec dataset categories with EfficientAD and FastFlow. $\Delta$ is the median improvement percentage of the guided result from baseline.

| Vision | | Time-series | |
|---|---|---|---|
| Baseline | Guided | Baseline | Guided |
| 126.50 | 236.37 | 10.86 | 12.49 |

Table 7: Comparison of baseline and guided generation median runtimes (seconds).

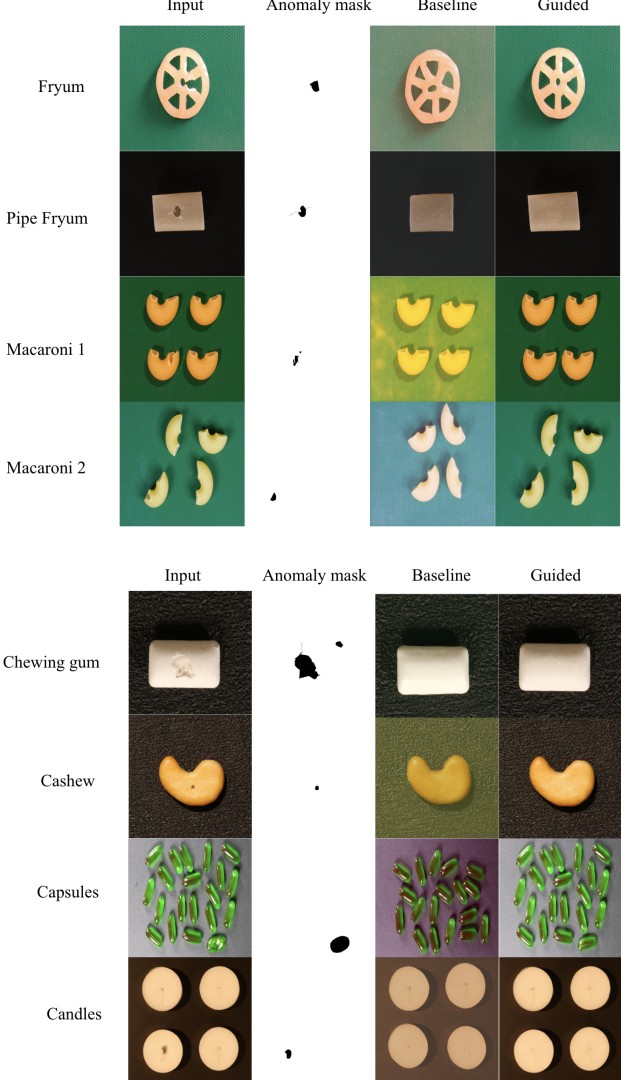

Figure 8: More VisA examples. With AR-Pro, we have anomaly repairs resemble inputs better, compared with baseline.

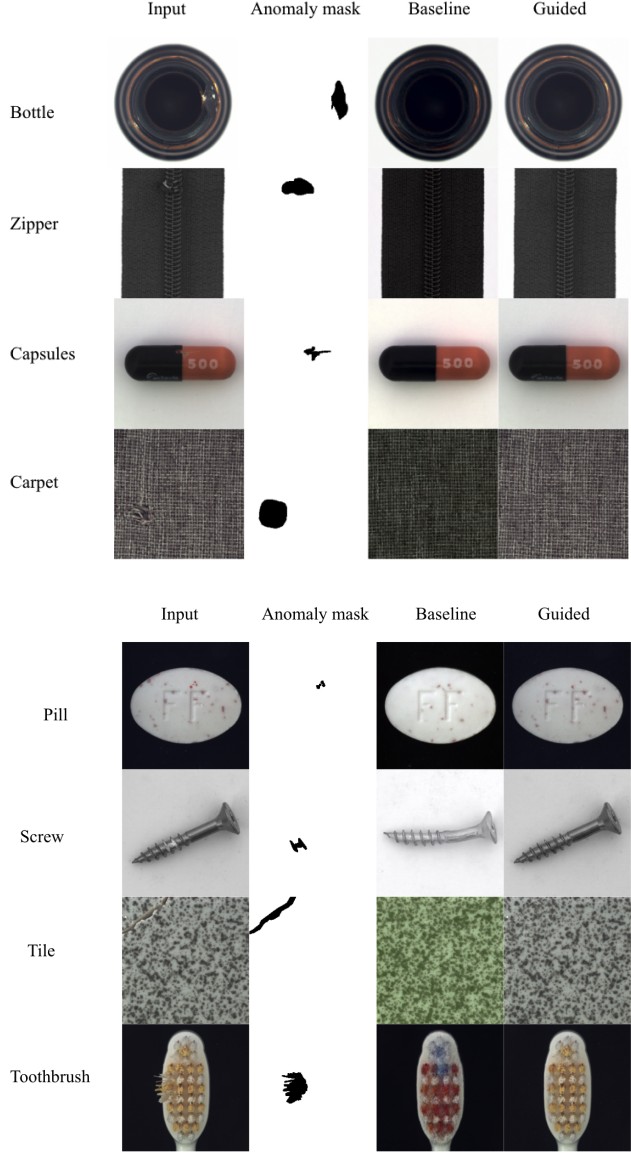

Figure 9: More MVTec examples (Part 1). With AR-Pro, anomaly repairs resemble inputs better compared with the baseline.

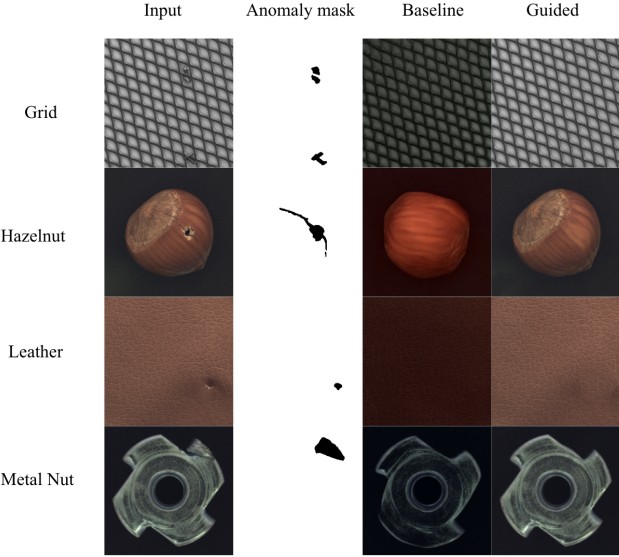

Figure 10: More MVTec examples (Part 2). With AR-Pro, anomaly repairs resemble inputs better compared with the baseline.

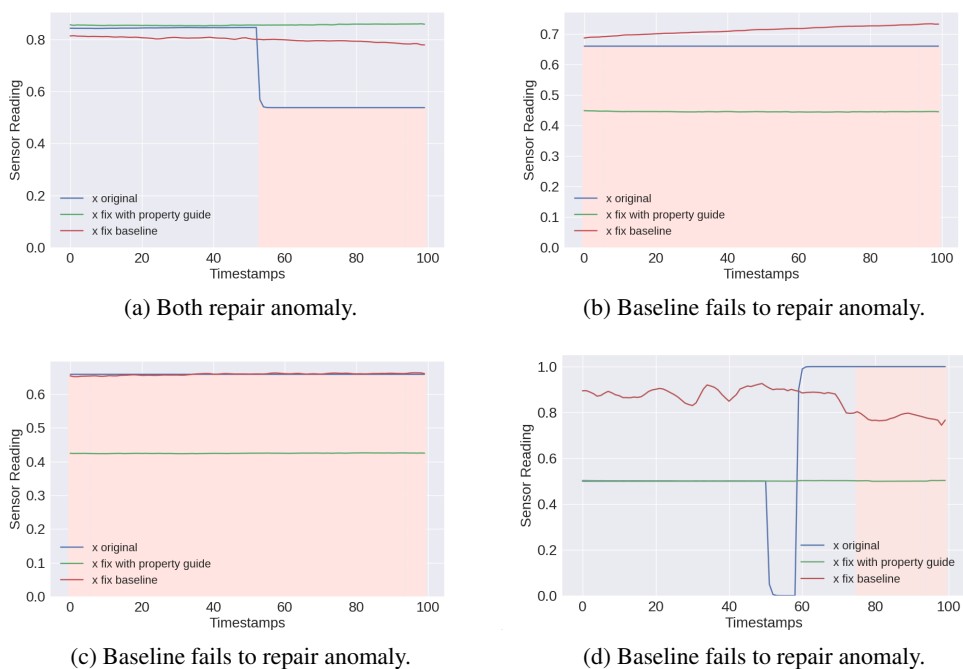

(a) Both repair anomaly.

(b) Baseline fails to repair anomaly.

(c) Baseline fails to repair anomaly.

(d) Baseline fails to repair anomaly.

Figure 11: The first image shows an instance where the baseline repairs the anomaly, but not as effectively as with property guidance. The last three images show instances where the baseline fails to repair the anomaly.

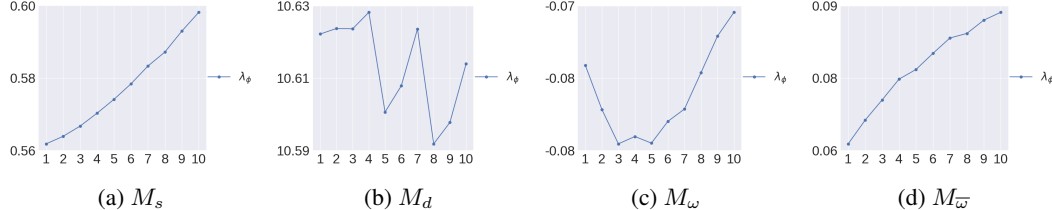

Figure 12: Effects of hyperparameter $\lambda_\phi$ on $M_s$, $M_d$, $M_\omega$ and $M_{\overline{\omega}}$ on SWaT; the effect of $\lambda_\phi$ varies across metrics, but the range remain relatively small.

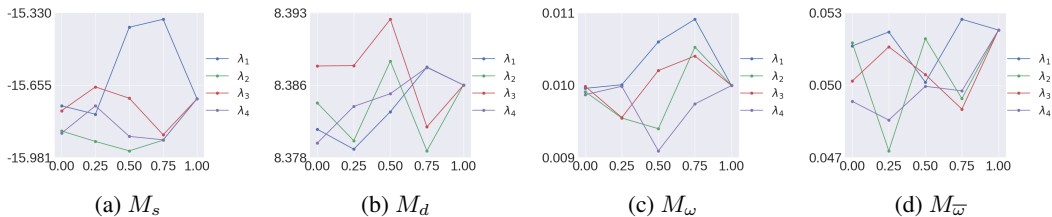

Figure 13: Effects of hyperparameter on $M_s$, $M_d$, $M_\omega$ and $M_{\overline{\omega}}$ on VisA cashew class; the effect of $\lambda_\phi$ varies across metrics, but the range remain relatively small.

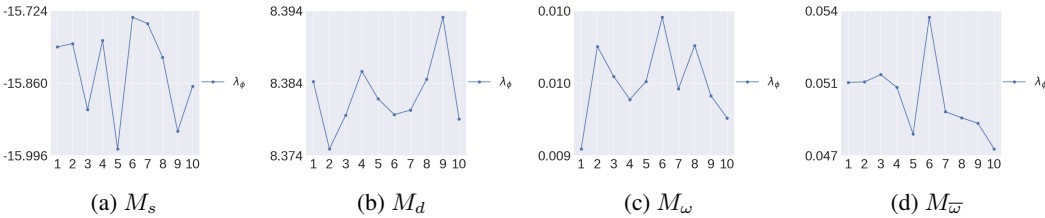

Figure 14: Effects of hyperparameter $\lambda_\phi$ on $M_s$, $M_d$, $M_\omega$ and $M_{\overline{\omega}}$ on VisA cashew class; the effect of $\lambda_\phi$ varies across metrics, but the range remain relatively small.

