# OpenReview forum: "AR-Pro: Counterfactual Explanations for Anomaly Repair with Formal Properties"
_NeurIPS.cc/2024/Conference — NeurIPS 2024 poster_

### Official Review · Reviewer_k8Af · 2024-06-27

**Soundness:** 3
**Presentation:** 3
**Contribution:** 3
**Rating:** 6
**Confidence:** 4

**Summary:**

The paper addresses the anomaly explanation task by repairing the normal appearances of input inputs. Specifically, this paper designs four properties to guide the repair process that works in both image and time series domains. To demonstrate the effectiveness of the proposed method, this paper conducts experiments on the VisA and SWaT datasets.

**Strengths:**

The motivation is clear, thus, repairing the normal appearances for better anomaly explanation.

The designed four properties to guide the repair process are reasonable.

The proposed metrics are reasonable and the proposed method achieves significant improvements versus the utilized baseline.

**Weaknesses:**

This paper exploits fixed anomaly detection methods, i.e., Fastflow and GPT2, to guide the anomaly repair process. The two selected methods are out-of-date, and the influence of the utilized anomaly detection methods should also be investigated.

More datasets should be included but not only VisA and SWaT. For example, for industrial image anomaly detection, MVTec AD should also be included.

For me, Property 1 is a combination of Property 3 and 4, and maybe only Property 2, 3, and 4 are enough.

The figure quality should be improved, especially for Figure 3.

For the fourth metric M_{1-w}, should it be the absolute value? Or it will encourage lower anomaly scores for normal regions in the fixed results.

Sec 3.3 is quite important but is not clear. The author may condense Sec. 2 a lot and then extend Sec 3.3, at least elaborating on the details of the generation process and adding proper references.


------------post rebuttal response--------
the authors have addressed all my concerns, so I raise my rate from 5 to 6.

**Questions:**

See the Weaknesses.

**Limitations:**

See the Weaknesses.

---

> ### Author Rebuttal · Authors · 2024-08-06
>
> We thank the reviewer for their useful suggestions on how to improve our experiments and presentation. We will incorporate these changes into our manuscript, and we believe that they will help us greatly improve the quality of our work. We have included some results of our work-in-progress experiments in the supplemental material, namely the addition of new models (EfficientAD [1] for vision, Llama-2-7b [2] for time-series) and datasets (MVTec [3] for vision, WADI [4] and HAI [5] for time-series). Below, we respond to the reviewer's comments and questions.
>
> * **Additional Models and Datasets.**
> We are in the process of incorporating more recent models (EfficientAD, Llama-2-7b) and datasets (MVTec, WADI, HAI) to strengthen our experiments. In particular, WADI [4] is a time-series dataset that focuses on water distribution systems and contains anomalies related to sensor malfunctions and system faults. HAI [5] is another time-series dataset used for industrial control systems and includes anomalies such as cyber-attacks and operational failures. Like SWaT, both WADI and HAI have ground-truth feature-level annotations of anomalies. Due to limits in computing resources, we only include a sample of the ongoing experiments in the supplemental PDF. Importantly, we observe similar trends as our other experiments.
>
>
> * **Relation Between Property 1 and Properties 3 and 4.**
> The reviewer is correct in noting the relation between Property 1 and Properties 3 and 4. After some deliberation, we felt that it was simpler for the exposition to keep them separate, particularly in their loss function encodings of Section 3.2. We tried some other formulations but could not reach a satisfactory balance of a smooth exposition and notational compactness. We are open to suggestions on how to rework the presentation, and we would be glad to hear if the reviewer might have some ideas.
>
> * **Figure 3 Quality.**
> We have provided an updated Figure 3 in our supplementary PDF.
>
> * **Metric of $M_{1-w}$.**
> We intended for this to be without the absolute value. Intuitively, the repair can have a lower anomaly score in the normal region since fixing the anomalous region may reduce the overall score of nearby features.
>
> * **Clarity of Section 3.3.**
> We thank the reviewer for identifying this weakness. We will revise this section for greater clarity and detail.
>
> [1] Batzner, Kilian, Lars Heckler, and Rebecca König. "Efficientad: Accurate visual anomaly detection at millisecond-level latencies." Proceedings of the IEEE/CVF Winter Conference on Applications of Computer Vision. 2024.
>
> [2] Touvron, Hugo, et al. "Llama 2: Open foundation and fine-tuned chat models." arXiv preprint arXiv:2307.09288 (2023).
>
> [3] Bergmann, Paul, et al. "MVTec AD--A comprehensive real-world dataset for unsupervised anomaly detection." Proceedings of the IEEE/CVF conference on computer vision and pattern recognition. 2019.
>
> [4] Ahmed, Chuadhry Mujeeb, Venkata Reddy Palleti, and Aditya P. Mathur. "WADI: a water distribution testbed for research in the design of secure cyber physical systems." Proceedings of the 3rd international workshop on cyber-physical systems for smart water networks. 2017.
>
> [5] Shin, Hyeok-Ki, et al. "{HAI} 1.0:{HIL-based} Augmented {ICS} Security Dataset." 13Th USENIX workshop on cyber security experimentation and test (CSET 20). 2020.

---

> > ### Comment · Reviewer_k8Af · 2024-08-07
> > **Response**
> >
> > Dear Authors,
> >
> > Thank you so much for your efforts. Explaining anomalies by repairing them is an interesting direction, and your responses have addressed my concerns.  I would suggest trying to repair semantic anomalies, like those anomalies in MVTec LOCO in the future since the structural anomalies are typically easy for users to understand the reason to be anomalous, but in comparison, users can suffer from explaining semantic anomalies without proper prior information.

---

> > > ### Author Response · Authors · 2024-08-07
> > > **Response to Reviewer k8Af**
> > >
> > > We thank the reviewer for their encouragement and dataset suggestions. We will try to incorporate MVTec LOCO into our paper. Meanwhile, if the reviewer has any additional comments, questions, or requests, please do let us know.

---

### Official Review · Reviewer_F7R5 · 2024-07-12

**Soundness:** 2
**Presentation:** 2
**Contribution:** 1
**Rating:** 3
**Confidence:** 4

**Summary:**

The paper proposes an anomaly repair technique. Based on proposed four
properties, it trains a generative model that can fix anomaly data to a benign
one. The proposed properties include similar to the data, etc., which are
globally applicable to any dataset. The evaluation is performed on VisA dataset
and the GPT2 on SWat Dataset. The results show that the proposed method can
effectively repair the anomaly data.

**Strengths:**

The paper can properly formulate the proposed properties and mix them into the
training of the generative model. The proposed method can effectively repair the
anomaly data. The paper is well-written and easy to follow.

**Weaknesses:**

* I do not get the motivation of fixing anomaly data. If it is detected and
  considered as anomaly, our typical action is to determine if it is a false
  positive and then improve the detection model and following downstream models.
  What is the rationale of fixing the anomaly data?

* Properties 3 and 4 seem to be generalized version of 1 and 2. Typically, this
  should be solved in the training of models, either a vision model or
  time-series. Why not directly embedding these properties into the training of
  final model instead of training a separate model?

* The method assumes the availability of anomaly map, which seems to be
  impractical in real-world scenarios. How can the method be applied to
  real-world scenarios where anomaly maps are not available? In the real world,
  we tend to only have individual anomaly data points, not a map. Similarly, it
  would be great if you can extend the discussion to the region selector.

**Questions:**

What is the motivation of fixing the anomaly data?

**Limitations:**

The paper does not explicitly discuss the limitations and potential negative. As
for me, my concern is this can be used as a tool to evade anomaly detection
models. The authors should discuss this in the paper.

---

> ### Author Rebuttal · Authors · 2024-08-06
>
> We thank the reviewer for their feedback on problem motivation and potential risks. We will revise our manuscript to address these concerns. Below are our responses to the comments and questions.
> * **Motivation for Fixing Anomalous Inputs.**
> Anomaly repair is useful when the input data is noisy and needs to be cleaned [1]. This is studied in the context of image data [2, 10], time-series signals [3,4], graph data [5], and also heterogeneous data [6]. A common application is to improve the quality of the training data, while another is to recover from distribution shifts and allow downstream tasks to operate on data that is more in-distribution, i.e. rainy conditions in autonomous driving [7], geolocation data [8]. There are different ways to perform the repair, for example, [9] repairs anomalies through semantic-preserving transformations on images, while our work uses diffusion models for images and time-series, guided by formal specifications.  In addition, we see a good opportunity to use repairs as an explainability method to improve the interpretability of black-box machine-learning models. Our proposed framework for counterfactual explanations is a step towards helping users better diagnose whether anomaly "hits" are indeed false positives by revealing what a non-anomalous input should have been like. Counterfactual explanations are especially relevant if the user is inexperienced or the data is complex [11].
> * **Properties 3 and 4 vs. Properties 1 and 2.**
> Although Properties 3 and 4 are similar to Properties 1 and 2, their respective loss function encodings (Section 3.2) are different. Namely, the loss for Property 1 may be negative, while those for Property 2, 3, and 4 may not be. Moreover, Property 2 concerns similarity, while Property 4 concerns the anomaly score. We will clarify the exposition around this part.
> * **Embedding Properties into Training.**
> We apologize for not fully understanding the reviewer's question, and we would appreciate some clarification. In the meantime, we hope the following can serve as a partial answer. We have previously tried to encode these properties into a repair model's training objective. In particular, we attempted to perform anomaly repair on image data with VAEs. Despite our best efforts, the repair models often produced blurry or incorrect outputs. It was only when we switched to iterative diffusion-style methods that we could attain high-quality repairs. We will expand our discussions on this.
> * **Availability of the Anomaly Map.**
> The availability of the anomaly map is indeed a concern, especially if the detector were proprietary or closed-source. For open-sourced models implemented with libraries like PyTorch, we found that the anomaly map was usually available. For instance, the anomaly map was available for many detectors in Anomalib [12]. Nevertheless, the reviewer raises a valid point that the availability of the anomaly map is dependent on software implementation and should not be assumed. We will update our manuscript to address this limitation.
> * **Region Selector**.
> Our anomalous region selector is the same as the commonly used thresholding methods. The user specifies a threshold vector $\tau \in \mathbb{R}^{n}$ to specify the anomaly threshold of each feature.
> * **Evasion of Anomaly Detectors.**
> Since our work uses the detector's anomaly score in an optimization objective, attackers could potentially misuse our techniques. We will update our manuscript to discuss these risks. However, if attacks on anomaly detectors become common, it would likely spur interest in developing robust detectors, similar to the advancements in adversarial image classification and LLM jailbreaking defenses that followed extensive study of defensing against attacks.
>
> [1] Yang, J., Zhou, K., Li, Y., & Liu, Z. (2024). Generalized out-of-distribution detection: A survey. International Journal of Computer Vision.
>
> [2] Eduardo, S. F. L. M. (2023). Data cleaning with variational autoencoders.
>
> [3] Wang, X., & Wang, C. (2019). Time series data cleaning: A survey. Ieee Access, 8, 1866-1881.
>
> [4] Zhang, A., Song, S., Wang, J., & Yu, P. S. (2017). Time series data cleaning: From anomaly detection to anomaly repairing. Proceedings of the VLDB Endowment, 10(10), 1046-1057.
>
> [5] Akoglu, L., Tong, H., & Koutra, D. (2015). Graph based anomaly detection and description: a survey. Data mining and knowledge discovery, 29, 626-688.
>
> [6] Eduardo, S., Nazábal, A., Williams, C. K., & Sutton, C. (2020, June). Robust variational autoencoders for outlier detection and repair of mixed-type data. In International Conference on Artificial Intelligence and Statistics. PMLR.
>
> [7] Filos, A., Tigkas, P., McAllister, R., Rhinehart, N., Levine, S., & Gal, Y. (2020, November). Can autonomous vehicles identify, recover from, and adapt to distribution shifts?. In International Conference on Machine Learning. PMLR.
>
> [8] Corizzo, R., Ceci, M., & Japkowicz, N. (2019). Anomaly detection and repair for accurate predictions in geo-distributed big data. Big Data Research.
>
> [9] Lin, V., Jang, K. J., Dutta, S., Caprio, M., Sokolsky, O., & Lee, I. (2024, June). DC4L: Distribution shift recovery via data-driven control for deep learning models. In 6th Annual Learning for Dynamics & Control Conference. PMLR.
>
> [10] Pirnay, Jonathan, and Keng Chai. "Inpainting transformer for anomaly detection." In International Conference on Image Analysis and Processing. Cham: Springer International Publishing, 2022.
>
> [11] Verma, Sahil, John Dickerson, and Keegan Hines. "Counterfactual explanations for machine learning: A review." arXiv preprint arXiv:2010.10596 2 (2020)
>
> [12] Akcay, Samet, Dick Ameln, Ashwin Vaidya, Barath Lakshmanan, Nilesh Ahuja, and Utku Genc. "Anomalib: A deep learning library for anomaly detection." In 2022 IEEE International Conference on Image Processing (ICIP). IEEE, 2022.

---

> > ### Comment · Reviewer_F7R5 · 2024-08-12
> > **Follow up**
> >
> > Could you explain why encoding the properties as training objectives fail? Thanks.

---

> > > ### Author Response · Authors · 2024-08-13
> > > **Explanation of why encoding as training objectives fail**
> > >
> > > Thank you for your response. We found that augmenting the training loss often failed to produce effective counterfactuals, either due to limitations in the model architecture or because of the model's natural training loss.
> > >
> > > After many attempts, we discovered that guided diffusion was a straightforward algorithm capable of achieving good performance. Our eventual choice of algorithm was primarily influenced by the available generative models for images (VAEs, GANs, Diffusion), and we elaborate on the challenges associated with each below.
> > >
> > > We initially tried VAEs for image repair but found the generated counterfactuals were often blurry, which is a known limitation of VAE [1,2]. Although later work describes methods for high-resolution VAE-based image generation [3,4], these methods often fell short in producing the same level of sharpness and detail compared to GANs and diffusion models—especially within the anomalous regions. So why are VAEs commonly used for reconstruction-based anomaly detection despite the blurry outputs? This is likely because although their ELBO loss tends to favor images that are the "average" of a distribution (i.e., blurry), this often suffices for detecting anomalous regions.
> > >
> > > Given the limitations of VAEs, we turned to GANs and diffusion models. However, GAN training was plagued by well-known issues of instability and non-convergence [5,6], thereby leading us to focus on diffusion models.
> > >
> > > With diffusion models, we found that incorporating the four properties into the training loss did not enhance counterfactual quality. This was likely because the diffusion model's noise prediction loss often led to static-like reconstructions (very noisy images) that were unsuitable for evaluation against our properties, especially at larger time steps. It was with guided diffusion methods [7] that we achieved high-quality counterfactuals, and these methods now form the foundation of our present methodology.
> > >
> > > We thank the reviewer for these questions, and we will update our manuscript to include a discussion of the lessons learned with various model architectures.
> > >
> > > **Additional References**
> > >
> > > [1] Tomczak, Jakub, and Max Welling. "VAE with a VampPrior." In International conference on artificial intelligence and statistics, pp. 1214-1223. PMLR, 2018.
> > >
> > > [2] Dai, Bin, and David Wipf. "Diagnosing and enhancing VAE models." arXiv preprint arXiv:1903.05789 (2019).
> > >
> > > [3] Razavi, Ali, Aaron Van den Oord, and Oriol Vinyals. "Generating diverse high-fidelity images with vq-vae-2." Advances in neural information processing systems 32 (2019).
> > >
> > > [4] Liu, Zhi-Song, Wan-Chi Siu, and Yui-Lam Chan. "Photo-realistic image super-resolution via variational autoencoders." IEEE Transactions on Circuits and Systems for video Technology 31, no. 4 (2020): 1351-1365.
> > >
> > > [5] Saxena, Divya, and Jiannong Cao. "Generative adversarial networks (GANs) challenges, solutions, and future directions." ACM Computing Surveys (CSUR) 54, no. 3 (2021): 1-42.
> > >
> > > [6] Lu, Yuzhen, Dong Chen, Ebenezer Olaniyi, and Yanbo Huang. "Generative adversarial networks (GANs) for image augmentation in agriculture: A systematic review." Computers and Electronics in Agriculture 200 (2022): 107208.
> > >
> > > [7] Dhariwal, Prafulla, and Alexander Nichol. "Diffusion models beat gans on image synthesis." Advances in neural information processing systems 34 (2021): 8780-8794.

---

### Official Review · Reviewer_DUtN · 2024-07-19

**Soundness:** 4
**Presentation:** 4
**Contribution:** 4
**Rating:** 7
**Confidence:** 4

**Summary:**

Paper proposes a method for anomaly repair that goes one step beyond an anomaly detection and/or an anomaly localization method. While anomaly detection focuses on identifying which objects (images, time series, etc.) are anomalous, and anomaly localization focuses on identifying regions within the object (image or time series) which is anomalous, anomaly repair focuses on producing the normal object that the anomalous object is derived from. Authors identify properties for the repair, and develop a generative model that can take an anomalous object and produces the corresponding normal (and repaired) object.

Paper describes experimental results to demonstrate the effectiveness of the proposed approach on different data sets. Evaluation is done both quantitatively and qualitatively.

**Strengths:**

Paper is well written except for some minor notational inconsistencies (see my questions). The idea is interesting and novel and targets an important and practical issue of anomaly repair. Experimental evaluation is robust and provides evidence regarding the effectiveness of the proposed method.

**Weaknesses:**

A primary weakness of this paper is that it does not state the assumptions regarding the scope of the methods upfront. The analysis holds for methods that follow the reconstruction-based anomaly detection recipe, i.e., each input is reconstructed, and the anomaly score is calculated using the difference between the input and reconstruction. While that is true for many methods, there is still a vast majority of methods for which this is not applicable. In fact, even in the reconstruction based methods, the analysis holds for those in which the scoring function can be decomposed over the individual features. Again, this is not true for all reconstruction-based methods. It would be good if the authors can make this clear in the beginning to avoid confusion.

**Questions:**

- In Section 2, the definition of anomaly map has a term $\hat{x}$, which has not been defined. From the figure 2 it appears that $\hat{x}$ is the reconstructed version of the original data point. Does that mean that this analysis framework is applicable only to the class of anomaly detection methods that involve reconstruction. While those methods are certainly capable, there is a large class of anomaly detection methods that do not necessarily have a reconstruction step involved. In fact, does the whole approach rely on availability of a base anomaly detector?
- Can this method be applied of multivariate instances (no spatial or temporal relationships). It is unclear how a single threshold ($\tau$) in line 106 be applied to a case where each feature could have a different scale.
- What is $\alpha_i(x)$ in Definition 2.1? Is it the absolute difference between the actual value and the reconstructed value for the $i^{th}$ feature?
- In Section 3.1, what does a "normal input" mean? Does it refer to an entire observation (an image) that does not have any anomalous features or does it refer to the non-anomalous parts of an image?
- In Figure 5(a), why do the authors infer that the proposed method is not adding a spurious signal, when the plot shows that it is different from the normal signal

**Limitations:**

I think the paper does not quite scope the method - identify what kind of problems will it work on and what it won't.

---

> ### Author Rebuttal · Authors · 2024-08-06
>
> We thank the reviewer for their positive review. Their feedback will help us greatly improve the clarity throughout the manuscript. We address the reviewer's comments and questions below.
>
> * **Scope of the assumptions.**
> Although we use reconstruction-based anomaly detection as a motivating example, we are not restricted to such methods. Rather, our framework focuses on anomaly detectors with a linearly decomposable score (Definition 2.1). Linear decomposability includes many reconstruction-based methods (e.g., VAEs) as well as maximum likelihood-based ones (e.g., FastFlow). Despite this, it does not cover all anomaly detectors, such as clustering-based methods. We will improve our manuscript to better discuss the scope and limitations of our work to avoid confusion.
>
> * **Definition of $\hat{x}$ in Section 2.**
> $\hat{x}$ denotes the reconstructed input obtained from reconstruction-based methods. We will update our manuscript to clarify this.
>
> * **Availability of base anomaly detector.**
> We assume that the base anomaly detector is available. This is because we take the anomaly score into consideration when performing repairs. We will update our manuscript to clarify this.
>
> * **Extension to multivariate instances.**
> Our framework can generalize to the multivariate case. For an input $x \in \mathbb{R}^n$, we use the $n$-dimensional thresholding vector $\tau \in \mathbb{R}^n$ to allow for fine-grained control of the anomaly region selector $\omega(x) \in \\{0,1\\}^n$ at each feature.
>
> * **$\alpha_i (x)$ in Definition 2.1.**
> $\alpha_i (x)$ is the $i$th coordinate value of the anomaly map. In the given example, it is the absolute value between the reconstruction and original image on the $i$th feature. We will update our exposition to clarify this.
>
>
> * **Definition of “normal input”.**
> We will update Section 3.1 of our manuscript to clarify that “normal input” means that the entire input (e.g., whole image) is considered non-anomalous.
>
> * **Spurious/normal signal in Figure 5(a).**
> This figure compares our method (green) with the baseline (red) and shows that we generate qualitatively better (less “spurious”) signals. We will improve the wording in the experiments section to avoid confusion for future readers.

---

> > ### Comment · Reviewer_DUtN · 2024-08-14
> >
> > Many thanks for your clarifications. I stand by my positive rating.

---

### Author Rebuttal · Authors · 2024-08-07

We thank the reviewers for their time and feedback. Their comments and suggestions will allow us to greatly improve our manuscript in exposition narrative, technical details, and experiment results. We are in the process of running additional experiments involving newer models like EfficientAD [1] and Llama-2 [2], as well as datasets like MVTec [3], WADI [4], and HAI [5]. We have attached some preliminary results in our supplemental PDF.


[1] Batzner, Kilian, Lars Heckler, and Rebecca König. "Efficientad: Accurate visual anomaly detection at millisecond-level latencies." Proceedings of the IEEE/CVF Winter Conference on Applications of Computer Vision. 2024.

[2] Touvron, Hugo, et al. "Llama 2: Open foundation and fine-tuned chat models." arXiv preprint arXiv:2307.09288 (2023).

[3] Bergmann, Paul, et al. "MVTec AD--A comprehensive real-world dataset for unsupervised anomaly detection." Proceedings of the IEEE/CVF conference on computer vision and pattern recognition. 2019.

[4] Ahmed, Chuadhry Mujeeb, Venkata Reddy Palleti, and Aditya P. Mathur. "WADI: a water distribution testbed for research in the design of secure cyber physical systems." Proceedings of the 3rd international workshop on cyber-physical systems for smart water networks. 2017.

[5] Shin, Hyeok-Ki, et al. "{HAI} 1.0:{HIL-based} Augmented {ICS} Security Dataset." 13Th USENIX workshop on cyber security experimentation and test (CSET 20). 2020.

---

### Decision · Program_Chairs · 2024-09-25

**Decision:**

Accept (poster)

**Comment:**

This paper proposed a method to convert anomalous to normal images under a set of properties such as making minimal changes to normal regions and actually reducing anomalous scores. The method uses diffusion methods for image AD, and also applies to time series anomalies. Overall, the method addressed an interesting problems and shows promising results. While there were some concerns, they were addressed in the rebuttal.

Reviewer DUtN was concerned the paper does not detail which AD segmentation approaches this method will work for. The authors explained this will not work for clustering based AD. The AC suggests also mentioning retrieval based AD e.g. SPADE or PatchCore. In either case the AC believes this can be fixed in one or two sentences in the final manuscript.

Reviewers k8Af and F7R5 pointed out there is some redundancy between the 4 formal conditions. The authors admitted that it is possible to remove condition 1, but it makes the exposition more natural. This satisfied reviewer k8Af. The AC believes this is reasonable but requests adding a clarification to the final version. Reviewers k8Af wanted additional experiments, the reviewers supplied a few experiments and promised further experiments, this addressed the concern.

Reviewer F7R5 initially had several additional concerns about i) the motivation for anomaly repair ii) that anomalous images do not come with anomaly maps. AC believes the concerns should not prevent acceptance as: i) fixing anomalies is interesting both as a way of obtaining pristine images and as a tool for explaining model decisions, which is very important for anomaly detector as they are essentially unsupervised  ii) there are many anomaly segmentation methods available and it is not essential to introduce another one. The reviewer also indicated post-rebuttal that they do not object to accepting this paper.

Overall, given the interesting research direction that this paper offers, the promising initial results and that rebuttal addressed the concerns of the reviewers, the AC recommends acceptance.